# Synthesis of 4,4′-(4-Formyl-1*H*-pyrazole-1,3-diyl)dibenzoic Acid Derivatives as Narrow Spectrum Antibiotics for the Potential Treatment of *Acinetobacter Baumannii* Infections

**DOI:** 10.3390/antibiotics9100650

**Published:** 2020-09-28

**Authors:** Evan Delancey, Devin Allison, Hansa Raj KC, David F. Gilmore, Todd Fite, Alexei G. Basnakian, Mohammad A. Alam

**Affiliations:** 1Department of Chemistry and Physics, College of Science and Mathematics, Arkansas State University, Jonesboro, AR 72467, USA; edelance@nyit.edu (E.D.); dra246@health.missouri.edu (D.A.); hansa.kc@smail.astate.edu (H.R.K.); 2Department of Biological Sciences, College of Science and Mathematics, Arkansas State University, Jonesboro, AR 72467, USA; dgilmore@astate.edu; 3Department of Pharmacology and Toxicology, University of Arkansas for Medical Sciences, 4301 W. Markham St., Little Rock, AR 72205, USA; todd.fite@va.edu (T.F.); basnakianalexeig@uams.edu (A.G.B.); 4Central Arkansas Veterans Healthcare System, W. 7th St., Little Rock, AR 72205, USA

**Keywords:** antimicrobial, pyrazole, hydrazone, *Acinetobacter baumannii*, toxicity, narrow-spectrum antibiotics

## Abstract

*Acinetobacter baumannii* has emerged as one of the most lethal drug-resistant bacteria in recent years. We report the synthesis and antimicrobial studies of 25 new pyrazole-derived hydrazones. Some of these molecules are potent and specific inhibitors of *A. baumannii* strains with a minimum inhibitory concentration (MIC) value as low as 0.78 µg/mL. These compounds are non-toxic to mammalian cell lines in in vitro studies. Furthermore, one of the potent molecules has been studied for possible in vivo toxicity in the mouse model and found to be non-toxic based on the effect on 14 physiological blood markers of organ injury.

## 1. Introduction

The ESKAPE pathogens, a group of six bacteria (*Enterococcus faecium*, *Staphylococcus aureus*, *Klebsiella pneumoniae*, *Acinetobacter baumannii*, *Pseudomonas aeruginosa*, and *Enterobacter species*) cause the majority of nosocomial infections in the U.S., and these pathogens are effectively escaping the current arsenal of antibiotics [1]. *A. baumannii,* one of the ESKAPE pathogens, is a commonly found aquatic Gram-negative bacterium, but its drug-resistant strains can cause serious health problems in immunocompromised individuals particularly in clinical settings. In just two decades, this bacterium became well-known to cause serious infections, being notorious opportunist and producing multiresistant strains in hospital wards and in communities [2,3,4,5]. *A. baumannii* infection of U.S. service members has become a major problem since the Iraq War began in 2003 [6]. In recent years, the emergence of *A. baumannii*, which is extremely drug- and pandrug-resistant, has alarmed the medical community [7,8,9]. In late February 2017, the WHO released a list of 12 drug-resistant bacteria that pose the greatest threat to human health and for which new antibiotics are desperately needed. Carbapenem-resistant *A. baumannii* is on the top of the list [10,11]. Therefore, the discovery of new antibacterial agents to treat *A. baumannii* infections is imperative. 

Although broad-spectrum antibiotics play a vital role in treating bacterial infections, there are some drawbacks to their use such as selection for and spread of resistance across multiple bacterial species and the detrimental effect upon the host microbiome. If the causative agent of the infection is known, the use of narrow-spectrum antibiotics can alleviate some of these problems. The development of narrow-spectrum antibiotics that do not cause cross-resistance in non-targeted microbes and elicit less collateral damage upon the host microbiome is an attractive approach to fight multidrug-resistance infections [12]. It is also expected that new drugs will be non-toxic in vivo at therapeutic doses. 

## 2. Results and Discussion

In our efforts to find potent antimicrobial agents [13], we have previously reported the synthesis of phenyl substituted pyrazole derivatives as potent *S. aureus* and *A. baumannii* growth inhibitors [14,15]. Fluoro-substitution on the phenyl ring increased the potency of molecules significantly [16,17]. Similarly, coumarin and naphthalene substituted compounds also showed potent activity against the tested strains, particularly *S. aureus* [18,19,20]. Our reported molecules are significantly lipophilic. To get less lipophilic compounds, we designed and synthesized a series of diylbenzoic acid derivatives of pyrazole to get compounds with better pharmacological properties (Figure 1). 

The reaction of hydrazinobenzoic acid (**1**) with 4-acetylbenzoic acid (**2**) formed the hydrazone (**3**), which on reaction with POCl_3_/*N*,*N*-dimethylformamide (DMF) afforded the diylbenzoic acid-derived pyrazole aldehyde (**4**) in 90% overall yield. The ease of synthesis of the pure aldehyde derivative (**4**) in a multigram scale without work-up or column purification helped to synthesize a series of hydrazone derivatives. The starting material was subjected to further reaction with several hydrazine derivatives to get novel hydrazone derivatives. We found the expected products (**5**–**29**) in a very good average yield (Scheme 1). 

These new compounds were tested against six bacterial strains: Two Gram-positive strains, *S. aureus* ATCC 25923 and *Bacillus subtilis* ATCC 6623, and four Gram-negative bacteria, *Enterobacter aerogenes* ATCC 13048, *Escherichia coli* ATCC 25922, *A. baumannii* ATCC 19606, type strain (Ab6), and *Pseudomonas aeruginosa* ATCC 27833. Remarkably, in initial screening tests, none of the compounds showed activity against the bacteria tested other than *A. baumannii* ATCC 19606 at concentrations less than 50 µg/mL. Because of the high specificity of compounds against *A. baumannii*, these molecules were further tested against two additional *A. baumannii* strains (Table 1). 

Products of aliphatic hydrazine derivatives (**5**, **6**, **7**, and **8**) did not show any activity against the tested bacterial strains. *N*,*N*-Disubstituted hydrazone derivatives (**9**, **10**, **11**, **12**, **13**, **14**, and **15**) also failed to show any noticeable antimicrobial property for the tested strains. *N*-Phenyl derivatives with halogen substitution in the phenyl ring exhibited potent activity against *A. baumannii*. 2-Fluorphenyl-susbstituted hydrazone (**16**) showed potent activity against the *A. baumannii* strains with MIC values as low as 3.125 µg/mL. 3-Fluorophenyl derivative (**17**) showed better activity against the *A. baumannii* strains with MIC values as low as 1.56 µg/mL. Chloro-substitution gave the most potent molecule (**18**) in the series. This molecule (**18**) inhibited the growth of *A. baumannii* ATCC 19606, type strain with an MIC value at submicrogram concentration. Two other strains, *A. baumannii* ATCC BAA-1605 and *A. baumannii* ATCC 747, were also inhibited efficiently by this novel molecule with minimum inhibitory concentration (MIC) values 3.125 and 1.56 µg/mL, respectively. The bromo-substituted compound (**19**) also showed potent activity against the tested *A. baumannii* strains with MIC value as low as 3.125 µg/mL. Dihalosubstitution, fluoro (**20**), and chloro (**21**), in the phenyl ring resulted in similar potency against the tested strains of *A. baumannii*. Mixed-halo substitutions, chloro, and fluoro (**22** and **23**), retained the potency of the compounds. Tetra-fluoro substitution decreased the potency of the resultant molecule (**24**) several-fold. Similarly, the pentafluoro compound (**25**) failed to inhibit the growth of the tested strains. Extremely electron withdrawing groups such as trifluoromethyl (**26**), cyano (**27**), carboxylic acid (**28**), and nitro (**29**) completely eliminated the potency of the molecules.

Based on the antimicrobial data, we could deduce a clear structure–activity relationship (SAR) of narrow spectrum antimicrobial agents. *N*,*N*-Disubstituted compounds failed to show any activity against the tested strains. Halogen-substituted *N*-phenyl hydrazone showed potent activity. Among these halogen-substituted compounds, mono-substitution showed the best result and the chlorine substitution gave the most potent compound (**18**). Increasing the number of halogen atoms in the phenyl ring decreased the activity of the molecules as no activity was observed for the penta-substituted compounds. 

### 2.1. Experimental Data 

**4,4′-(4-formyl-1*H*-pyrazole-1,3-diyl)dibenzoic acid (4).** Yellow solid (3.02 g 90%). ^1^H NMR, 300 MHz (DMSO-*d*_6_): δ 9.99 (s, 1H), 9.41 (s, 1H), 8.09–8.05 (m, 8H); ^13^C NMR (75 MHz, DMSO-*d*_6_) δ = 189.6, 172.1, 171.6, 156.9, 146.5, 141.1, 140.2, 136.1, 134.6, 133.9, 127.9, 124.0. HRMS (ESI-FTMS Mass (m/z): calcd for C_18_H_12_N_2_O_5_ [M + H]^+^ = 337.0819, found 337.0821.

**4-[1-(4-carboxyphenyl)-4-[(*E*)-(dimethylhydrazono)methyl]pyrazol-3-yl]benzoic acid (5).** Yellow solid (347 mg, 92%). ^1^H NMR (300 MHz DMSO-*d*_6_): 8.77 (s, 1H), 8.11–8.04 (m, 6H), 7.90 (d, *J* = 8.2 Hz, 2H), 7.29 (s, 1H), 2.88 (s, 6H); ^13^C NMR (75 MHz DMSO-*d*_6_): 167.5, 167.1, 149.9, 142.5, 137.2, 131.3, 130.7, 130.1, 128.8, 128.5, 126.4, 124.2, 120.9, 118.4, 42.8. HRMS (ESI-FTMS Mass (m/z): calcd for C_20_H_18_N_4_O_4_ [M + H]^+^ = 379.1401, found 379.1398.

**4-[1-(4-carboxyphenyl)-4-[(*E*)-1-piperidyliminomethyl]pyrazol-3-yl]benzoic acid (6).** Yellow solid, (376 mg, 91%). ^1^H NMR 300 MHz (DMSO-*d*_6_): 8.77 (s, 1H), 8.06–8.03 (m, 6H), 7.88 (d, *J* = 8.0 Hz, 2H), 7.59 (s, 1H), 3.05 (s, 4H), 1.63 (s, 4H), 1.46 (s, 2H); ^13^C NMR (75 MHz, DMSO-*d*_6_) δ = 167.6, 167.2, 150.2, 142.4, 137.0, 131.3, 131.0, 130.0, 129.2, 128.6, 126.8, 126.1, 120.7, 118.4, 51.8, 25.0, 24.0. HRMS (ESI-FTMS Mass (m/z): calcd for C_23_H_23_N_5_O_4_ [M + H]^+^ = 434.1823, found 434.1819.

**4,4′-{4-[(*E*)-(4-methylpiperazin-1-yl)imino)methyl]-1*H*-pyrazole-1,3-diyl}dibenzoic acid (7).** Yellow solid (402 mg, 93%). ^1^H NMR 300 MHz (DMSO-*d*_6_): δ 8.80 (s, 1H), 8.06 (s, 4H), 8.04 (d, *J* = 8.7 Hz, 2H), 7.88 (d, *J* = 8.3 Hz, 2H), 7.64 (s, 1H), 3.13 (br s, 4H), 2.60 (br s, 4H), 2.29 (s, 3H); ^13^C NMR (75 MHz, DMSO-*d*_6_ + CDCl_3_) δ = 167.8, 167.4, 150.3, 142.3, 136.7, 131.4, 131.3, 130.0, 129.7, 128.6, 128.0, 127.1, 120.3, 118.5, 53.9, 50.4, 45.4. HRMS (ESI-FTMS Mass (m/z): calcd for C_23_H_23_N_5_O_4_ [M + H]^+^ = 434.1823, found 434.1819.

**4-[1-(4-carboxyphenyl)-4-[(*E*)-(4-cyclopentylpiperazin-1-yl)iminomethyl]pyrazol-3-yl]benzoic acid (8).** Yellow solid (457 mg, 94%). ^1^H NMR, 300 MHz (DMSO-*d*_6_): 8.83 (s, 1H), 8.11–8.04 (m, 6H), 7.89 (d, *J* = 8. 1 Hz, 2H), 7.62 (s, 1H), 3.08 (s, 4H), 2.60 (s, 4H), 2.50 (s, 1H), 1.79 (s, 2H), 1.59–1.34 (m, 6H), ^13^C NMR (75 MHz, DMSO-*d*_6_) δ = 167.6, 167.1, 150.3, 142.4, 136.9, 131.3, 131.0, 130.1, 129.3, 128.6, 127.5, 127.0, 120.4, 118.5, 66.8, 51.2, 51.0, 30.3, 24.1. HRMS (ESI-FTMS Mass (m/z): calcd for C_27_H_29_N_5_O_4_ [M + H]^+^ = 488.2292, found 488.2301.

**4-[1-(4-carboxyphenyl)-4-[(*E*)-(diphenylhydrazono)methyl]pyrazol-3-yl]benzoic acid (9).** Yellow solid (446 mg, 89%). ^1^H NMR, 300 MHz (DMSO-*d*_6_): δ 9.12 (s, 1H), 8.14–8.06 (m, 4H), 7.97 (d, *J* = 8.1 Hz, 2H), 7.70 (d, *J* = 8.1 Hz, 2H), 7.44 (t, *J* = 7.8 Hz, 4H), 7.28–7.14 (m, 7H); ^13^C NMR (75 MHz, DMSO-*d*_6_) δ = 167.4, 167.0, 150.5, 143.2, 142.4, 136.9, 131.3, 130.9, 130.4, 129.8, 129.1, 128.6, 128.0, 127.8, 125.0, 122.5, 119.5, 118.5. HRMS (ESI-FTMS Mass (m/z): calcd for C_30_H_22_N_4_O_4_ [M + H]^+^ = 503.1714, found 503.1713.

**4-[4-[(*E*)-[benzyl(phenyl)hydrazono]methyl]-1-(4-carboxyphenyl)pyrazol-3-yl]benzoic acid (10).** Yellow solid (464 mg, 90%). (^1^H NMR, 300 MHz (DMSO-*d*_6_): δ 9.04 (s, 1H), 8.10–8.07 (m, 4H), 7.87 (d, *J* = 8.0 Hz, 2H), 7.47–7.35 (m, 9H), 7.30 (t, *J* = 7.9 Hz, 1H), 7.21 (d, *J* = 7.1 Hz, 2H), 6.91 (t, *J* = 7.2 Hz, 1H), 5.30 (s, 2H); ^13^C NMR (75 MHz, DMSO-*d*_6_) δ = 168.0, 167.7, 150.3, 147.7, 142.0, 136.5, 136.1, 132.5, 131.2, 130.0, 129.5, 129.3, 127.9, 127.5, 126.8, 126.6, 125.8, 120.5, 120.0, 118.4, 114.7, 48.9. HRMS (ESI-FTMS Mass (m/z): calcd for C_31_H_24_N_4_O_4_ [M + H]^+^ = 517.1870, found 517.1878.

**4-[1-(4-carboxyphenyl)-4-[(*E*)-(dibenzylhydrazono)methyl]pyrazol-3-yl]benzoic acid (11).** Yellow solid (487 mg, 92%). ^1^H NMR, 300 MHz (DMSO-*d*_6_): 8.79 (s, 1H), 8.11–8.03 (m, 4H), 7.82 (d, *J* = 8.1 Hz, 2H), 7.39–7.21 (m, 12H), 7.08 (s, 1H), 4.56 (s, 4H); ^13^C NMR (75 MHz DMSO-*d*_6_): 167.4, 167.1, 149.6, 142.5, 138.3, 136.8, 131.3, 130.4, 129.9, 129.0, 128.6, 127.9, 127.8, 127.5, 125.9, 123.9, 120.6, 118.5, 58.1. HRMS (ESI-FTMS Mass (m/z): calcd for C_32_H_26_N_4_O_4_ [M + H]^+^ = 531.2027, found 531.2031.

**4-[1-(4-carboxyphenyl)-4-[(*E*)-[ethyl(phenyl)hydrazono]methyl]pyrazol-3-yl]benzoic acid (12).** Yellow solid (399 mg, 88%). ^1^H NMR, 300 MHz (DMSO-*d*_6_): 9.04 (s, 1H), 8.17–8.07 (m, 6H), 7.91 (m, 2H), 7.69 (s, 1H), 7.32–7.21 (m, 4H), 6.85 (t, 1H), 4.02 (q, *J* = 6.5 Hz, 2H), 1.13 (t, *J* = 6.6 Hz, 3H); ^13^C NMR (75 MHz DMSO-*d*_6_): 167.5, 167.1, 150.4, 146.5, 142.5, 137.3, 131.3, 130.8, 130.1, 129.4, 128.9, 128.8, 127.3, 123.7, 120.8, 120.1, 118.5, 114.4, 38.7, 9.9. HRMS (ESI-FTMS Mass (m/z): calcd for C_26_H_22_N_4_O_4_ [M + H]^+^ = 455.1714, found 455.1720.

**4-[4-[(*E*)-[butyl(phenyl)hydrazono]methyl]-1-(4-carboxyphenyl)pyrazol-3-yl]benzoic acid (13).** Yellow solid (428 mg, 89%). ^1^H NMR, 300 MHz (DMSO-*d*_6_): 9.06 (s, 1H), 8.17–8.05 (m, 6H), 7.89 (d, *J* = 8.2 Hz, 2H), 7.65 (s, 1H), 7.33–7.21 (m, 4H), 6.84 (t, *J* = 6.9 Hz, 1H), 3.92 (s, 2H), 1.55–1.53 (m, 2H), 1.42–1.34 (m, 2H), 0.93 (t, *J* = 7.1 Hz, 3H); ^13^C NMR (75 MHz DMSO-*d*_6_): 167.5, 167.1, 150.4, 146.9, 142.5, 137.2, 131.3, 130.8, 130.0, 129.3, 128.9, 128.7, 127.1, 123.7, 120.7, 120.1, 118.5, 114.4, 43.9, 26.4, 20.0, 14.2. HRMS (ESI-FTMS Mass (m/z): calcd for C_28_H_26_N_4_O_4_ [M + H]^+^ = 483.2027, found 483.2027.

**4-[1-(4-carboxyphenyl)-4-[(*E*)-[methyl(m-tolyl)hydrazono]methyl]pyrazol-3-yl]benzoic acid (14).** Yellow solid (408 mg, 90%). ^1^H NMR (300 MHz DMSO-*d*_6_): δ 9.00 (s, 1H), 8.15–8.06 (m, 6H), 7.94 (d, *J* = 8.1 Hz, 2H), 7.64 (s, 1H), 7.13–7.09 (m, 3H), 6.66 (s, 1H), 3.37 (s, 3H), 2.25 (s, 3H); ^13^C NMR (75 MHz, DMSO-*d*_6_) δ = 167.5, 167.1, 150.3, 147.6, 142.5, 138.4, 137.4, 131.4, 130.7, 130.1, 129.1, 128.9, 128.8, 127.6, 124.3, 121.2, 120.7, 118.5, 115.5, 112.2, 32.9, 21.9. HRMS (ESI-FTMS Mass (m/z): calcd for C_26_H_22_N_4_O_4_ [M + H]^+^ = 455.1714, found 455.1720.

**4-[1-(4-carboxyphenyl)-4-[(*E*)-[ethyl(*p*-tolyl)hydrazono]methyl]pyrazol-3-yl]benzoic acid (15).** Yellow solid (411 mg, 88%). ^1^H NMR, 300 MHz (DMSO-*d*_6_): δ 9.01 (s, 1H), 8.18–8.06 (m, 6H), 7.90 (d, *J* = 8.1 Hz, 2H), 7.63 (s, 1H), 7.20 (d, *J* = 8.3 Hz, 2H), 7.05 (d, *J* = 8.3 Hz, 2H), 3.98 (q, *J* = 6.4 Hz, 2H), 2.23 (s, 3H), 1.11 (t, *J* = 6.3 Hz, 3H); ^13^C NMR (75 MHz, DMSO-*d*_6_) δ = 167.5, 167.1, 150.3, 144.4, 142.6, 137.3, 131.3, 130.7, 130.1, 129.8, 128.9, 128.8, 128.7, 127.2, 123.0, 120.9, 118.5, 114.6, 20.5, 9.9. HRMS (ESI-FTMS Mass (m/z): calcd for C_27_H_24_N_4_O_4_ [M + H]^+^ = 469.1870, found 469.1872.

**4,4′-(4-{(*E*)-[2-(2-fluorophenyl)hydrazinylidene]methyl}-1*H*-pyrazole-1,3-diyl)dibenzoic acid (16).** Brownish solid (346 mg, 78%). ^1^H NMR, 300 MHz (DMSO-*d*_6_): δ 10.52 (s, 1H), 9.12 (s, 1H), 8.16–8.07 (m, 6H), 7.99 (s, 1H), 7.90 (d, *J* = 8.1 Hz, 2H), 7.22–7.15 (m, 1H), 6.86 (d, *J* = 11.7 Hz, 1H), 6.73 (d, *J* = 8.1 Hz, 1H), 6.49 (t, *J* = 6.7 Hz, 1H); ^13^C NMR (75 MHz, DMSO-*d*_6_) δ = 167.5, 167.1, 163.8 (^1^*J*_C-F_ = 238.8 Hz), 150.4, 147.7 (^3^*J*_C-F_ = 11.1 Hz), 142.4, 136.9, 131.3, 131.03 (d, ^3^*J*_C-F_ = 10.0 Hz), 131.0, 130.1, 130.0, 129.1, 128.8, 127.7, 119.6, 118.6, 108.4, 104.5 (^3^*J*_C-F_ = 21.2 Hz), 98.9 (^3^*J*_C-F_ = 26.0 Hz). HRMS (ESI-FTMS Mass (m/z): calcd for C_24_H_17_FN_4_O_4_ [M + H]^+^ = 445.1307, found 445.1309.

**4,4′-(4-{(*E*)-[2-(3-fluorophenyl)hydrazinylidene]methyl}-1*H*-pyrazole-1,3-diyl)dibenzoic acid (17).** Brownish solid (346 mg, 78%). ^1^H NMR, 300 MHz (DMSO-*d*_6_): δ 10.47 (s, 1H), 9.08 (s, 1H), 8.15–7.90 (m, 4H), 7.97 (s, 1H), 7.75 (d, *J* = 7.4 Hz, 2H), 7.55–7.48 (m, 2H), 7.19 (q, *J* = 6.8 Hz, 1H), 6.88 (d, *J* = 11.7 Hz, 1H), 6.74 (d, *J* = 8.0 Hz, 1H), 6.49 (t, *J* = 8.3 Hz, 1H); ^13^C NMR (75 MHz, DMSO-*d*_6_) δ = 167.1, 163.9 (d, ^1^*J* =238.6 Hz), 151.6, 147.8 (d, ^3^*J* = 11.1 Hz), 142.6, 132.7, 131.3, 131.0 (d, ^3^*J* = 10.1 Hz), 130.4, 129.0, 128.8, 127.2, 119.2, 118.9, 118.4, 108.4, 105.0 (^2^*J* = 20.5), 104.7, 98.8 (d, 2*J* = 26.1 Hz). HRMS (ESI-FTMS Mass (m/z): calcd for C_24_H_17_FN_4_O_4_ [M + H]^+^ = 445.1307, found 445.1310.

**4,4′-(4-{(*E*)-[2-(4-chlorophenyl)hydrazinylidene]methyl}-1*H*-pyrazole-1,3-diyl)dibenzoic acid (18).** Brownish solid (349 mg, 76%). ^1^H NMR, 300 MHz (DMSO-*d*_6_): δ 10.42 (s, 1H), 9.07 (s, 1H), 8.16–8.07 (m, 6H), 7.98 (s, 1H), 7.90 (d, *J* = 8.2 Hz, 2H), 7.22 (d, *J* = 8.7 Hz, 2H), 7.02 (d, *J* = 8.7 Hz, 2H); ^13^C NMR (75 MHz, DMSO-*d*_6_) δ = 167.5, 167.1, 150.4, 144.6, 142.4, 136.9, 131.3, 131.0, 130.0, 129.8, 129.7, 128.8, 127.5, 122.1, 119.7, 118.6, 113.6. HRMS (ESI-FTMS Mass (m/z): calcd for C_24_H_17_ClN_4_O_4_ [M + H]^+^ = 461.1011, found 461.1008.

**4,4′-(4-{(*E*)-[2-(3-bromophenyl)hydrazinylidene]methyl}-1*H*-pyrazole-1,3-diyl)dibenzoic acid (19).** Brownish solid (388 mg, 77%). ^1^H NMR, 300 MHz (DMSO-*d*_6_): δ 10.47 (s, 1H), 9.22 (s, 1H), 8.15–8.08 (m, 2H), 7.99 (s, 1H), 7.91 (d, *J* = 8.3 Hz, 2H), 7.21–7.20 (m, 1H), 7.11 (t, *J* = 7.9 Hz, 1H), 6.91–6.84 (m, 2H); ^13^C NMR (75 MHz, DMSO-*d*_6_) δ = 167.5, 167.1, 150.5, 147.2, 142.4, 136.9, 131.3, 131.1, 130.9, 130.5, 130.0, 129.3, 128.9, 127.9, 123.0, 121.2, 119.5, 118.6, 114.3, 111.3. HRMS (ESI-FTMS Mass (m/z): calcd for C_24_H_17_BrN_4_O_4_ [M + H]^+^ = 505.0506, found 505.0512.

**4,4′-(4-{(*E*)-[2-(2,5-difluorophenyl)hydrazinylidene]methyl}-1*H*-pyrazole-1,3-diyl)dibenzoic acid (20).** Brownish solid (337 mg, 73%). ^1^H NMR, 300 MHz (DMSO-*d*_6_): δ 10.42 (s, 1H), 9.18 (s, 1H), 8.33 (s, 1H), 8.17–8.07 (m, 6H), 7.89 (d, *J* = 8.2 Hz, 2H), 7.25–7.10 (m, 2H), 6.52–6.47 (m, 1H); ^13^C NMR (75 MHz, DMSO-*d*_6_) δ = 167.5, 167.1, 160.1 (^1^*J*_C-F_ = 236.4 Hz), 150.6, 145.5 (d, *J* = 234.9 Hz), 142.4, 136.8, 135.3-135.1 (m), 133.2, 131.3, 131.1, 130.0, 129.3, 128.8, 127.9, 119.4, 118.6, 118.6-116.1 (m), 104.0 (^2,3^*J*_C-F_ = 25.0, 7.5 Hz), 100.8 (^2^*J*_C-F_ = 30.4 Hz). HRMS (ESI-FTMS Mass (m/z): calcd for C_24_H_16_F_2_N_4_O_4_ [M + H]^+^ = 463.1212, found 463.1214.

**4,4′-(4-{(*E*)-[2-(2,4-dichlorophenyl)hydrazinylidene]methyl}-1*H*-pyrazole-1,3-diyl)dibenzoic acid (21).** Brownish solid (376 mg, 76%). ^1^H NMR, 300 MHz (DMSO-*d*_6_): δ 10.04 (s, 1H), 9.11 (s, 1H), 8.47 (s, 1H), 8.16–8.08 (m, 6H), 7.88 (d, *J* = 8.1 Hz, 2H), 7.54 (d, *J* = 8.8 Hz, 1H), 7.43 (d, *J* = 2.0 Hz, 1H), 7.25 (d, *J* = 8.8 Hz, 1H); ^13^C NMR (75 MHz, DMSO-*d*_6_) δ = 167.5, 167.1, 150.7, 142.4, 141.0, 136.7, 133.7, 131.3, 131.1, 130.0, 129.3, 128.9, 128.7, 128.2, 127.6, 122.3, 119.5, 118.6, 116.7, 115.3. HRMS (ESI-FTMS Mass (m/z): calcd for C_24_H_16_Cl_2_N_4_O_4_ [M + H]^+^ = 495.0621, found 495.0618.

**4-[1-(4-carboxyphenyl)-4-[(*E*)-[(3-chloro-2-fluoro-phenyl)hydrazono]methyl]pyrazol-3-yl]benzoic acid (22).** Brownish solid (358 mg, 75%). ^1^H NMR, 300 MHz (DMSO-*d*_6_): δ 10.40 (s, 1H), 9.09 (s, 1H), 8.32 (s, 1H), 8.11–8.07 (m, 6H), 7.88 (s, *J* = 7.8 Hz, 2H), 7.43 (t, *J* = 7.6 Hz, 1H), 7.05 (t, *J* = 7.7 Hz, 1H), 6.85 (t, *J* = 6.4 Hz, 1H); ^13^C NMR (75 MHz, DMSO-*d*_6_) δ = 167.5, 167.1, 150.6, 144.7 (d, ^1^*J* = 235.8 Hz), 142.4, 136.8, 135.3 (d, ^3^*J* = 9.2), 133.2, 131.3, 131.1, 130.0, 129.3, 128.8, 127.7, 125.8 (d, *J* = 3.5 Hz), 119.9 (d, ^2^*J* = 14.2 Hz), 119.4, 118.8, 118.6, 113.0. HRMS (ESI-FTMS Mass (m/z): calcd for C_24_H_16_ClFN_4_O_4_ [M + H]^+^ = 479.0917, found 479.0920.

**4,4′-(4-{(*E*)-[2-(3-chloro-4-fluorophenyl)hydrazinylidene]methyl}-1*H*-pyrazole-1,3-diyl)dibenzoic acid (23).** Brownish solid (368 mg, 77%). ^1^H NMR, 300 MHz (DMSO-*d*_6_): δ 10.83 (br s, 1H), 9.05 (s, 1H), 8.10–8.98 (m, 7H), 7.81 (d, *J* = 8.1 Hz, 2H), 7.25–7.19 (m, 2H), 6.98–6.94 (m, 1H); ^13^C NMR (75 MHz, DMSO-*d*_6_) δ = 168.8, 168.4, 150.7, 151.1 (^1^*J*_C-F_ = 234.6 Hz), 150.7, 143.4, 141.2, 135.8, 135.1, 134.0, 131.0, 129.9, 128.2, 127.1, 120.3 (^2^*J*_C-F_ = 18.2 Hz), 119.2, 118.1, 117.6 (^2^*J*_C-F_ = 22.1 Hz), 112.6, 111.9. HRMS (ESI-FTMS Mass (m/z): calcd for C_24_H_16_ClFN_4_O_4_ [M + H]^+^ = 479.0917, found 479.0919.

**4,4′-(4-{(*E*)-[2-(2,3,5,6-tetrafluorophenyl)hydrazinylidene]methyl}-1*H*-pyrazole-1,3-diyl)dibenzoic acid (24).** Brownish solid (398 mg, 80%). ^1^H NMR, 300 MHz (DMSO-*d*_6_): δ 10.38 (s, 1H), 8.81 (s, 1H), 8.36 (s, 1H), 8.05–7.98 (m, 6H), 7.81 (d, *J* = 6.9 Hz, 2H), 7.25–7.13 (m, 1H); ^13^C NMR (75 MHz, DMSO-*d*_6_) δ = 168.4, 168.1, 157.5, 151.0, 140.9, 138.5, 136.1, 135.8, 134.7, 131.0, 129.8, 128.2, 127.2, 118.3, 95.7 (^2^*J*_C-F_ = 25.0 Hz). HRMS (ESI-FTMS Mass (m/z): calcd for C_24_H_14_F_4_N_4_O_4_ [M + H]^+^ = 499.1024, found 499.1021.

**4,4′-(4-{(*E*)-[2-(2,3,4,5,6-pentafluorophenyl)hydrazinylidene]methyl}-1*H*-pyrazole-1,3-diyl)dibenzoic acid (25).** Brownish solid (423 mg, 82%). ^1^H NMR, 300 MHz (DMSO-*d*_6_): δ 10.13 (br s, 1H), 8.72 (s, 1H), 8.21 (s, 1H), 8.01 (s, 7H), 7.84–7.81 (m, 2H); ^13^C NMR (75 MHz, DMSO-*d*_6_) δ = 167.6, 167.2, 150.5, 142.1, 139.9–139.7 (m), 138.9, 136.4, 135.5, 135.3, 131.2, 129.8, 129.6–129.2 (m), 129.4, 128.6, 127.4, 121.6, 118.7, 118.5. HRMS (ESI-FTMS Mass (m/z): calcd for C_24_H_13_F_5_N_4_O_4_ [M + H]^+^ = 517.0930, found 517.0932.

**4,4′-(4-{(*E*)-[2-(4-trifluoromethylphenyl)hydrazinylidene]methyl}-1*H*-pyrazole-1,3-diyl)dibenzoic acid (26).** Brownish solid (370 mg, 75%). ^1^H NMR, 300 MHz (DMSO-*d*_6_): δ 10.78 (s, 1H), 9.07 (s, 1H), 8.14–8.07 (m, 8H), 7.90 (d, *J* = 8.3 Hz, 2H), 7.48 (d, *J* = 8.6 Hz, 2H), 7.13 (d, *J* = 8.5 Hz, 2H); ^13^C NMR (75 MHz, DMSO-*d*_6_) δ = 167.5, 167.1, 150.6, 148.6, 142.4, 136.8, 131.5, 131.3, 131.0, 130.0, 129.1, 128.8, 127.8, 126.7, 123.6, 119.4, 118.6, 111.8. HRMS (ESI-FTMS Mass (m/z): calcd for C_25_H_17_F_3_N_4_O_4_ [M + H]^+^ = 495.1275, found 495.1272.

**4-[1-(4-carboxyphenyl)-4-[(*E*)-[(4-cyanophenyl)hydrazono]methyl]pyrazol-3-yl]benzoic acid (27).** Brownish solid (342 mg, 76%). ^1^H NMR, 300 MHz (DMSO-*d*_6_): δ 10.92 (s, 1H), 9.12 (s, 1H), 8.16–8.07 (m, 8H), 7.90 (d, *J* = 8.3 Hz, 2H), 7.59 (d, *J* = 8.7 Hz, 2H), 7.10 (d, *J* = 8.6 Hz, 2H); ^13^C NMR (75 MHz, DMSO-*d*_6_) δ = 167.4, 167.0, 150.7, 148.9, 142.4, 136.8, 134.0, 132.6, 131.4, 131.1, 130.0, 129.2, 128.8, 128.0, 120.5, 119.2, 118.7, 112.4, 99.5. HRMS (ESI-FTMS Mass (m/z): calcd for C_25_H_17_N_5_O_4_ [M + H]^+^ = 452.1353, found 452.1360.

**4,4′-(4-{(*E*)-[2-(4-corboxyphenyl)hydrazinylidene]methyl}-1*H*-pyrazole-1,3-diyl)dibenzoic acid (28).** Brownish solid (347 mg, 74%). ^1^H NMR, 300 MHz (DMSO-*d*_6_): δ 12.8 (br s, 3H), 10.77 (s, 1H), 9.12 (s, 1H), 8.17–8.07 (m, 8H), 7.91 (s, 1H), 7.79–7.77 (d, *J* = 8.58 Hz, 2H), 7.05 (d, *J* = 8.4 Hz, 2H); ^13^C NMR (75 MHz, DMSO-*d*_6_) δ = 167.7, 167.5, 167.1, 150.6, 149.1, 142.4, 136.9, 131.5, 131.4, 131.0, 130.0, 129.9, 129.1, 129.0, 128.9, 120.5, 119.4, 118.6, 111.4. HRMS (ESI-FTMS Mass (m/z): calcd for C_25_H_18_N_4_O_6_ [M + H]^+^ = 471.1299, found 471.1299.

**4,4′-(4-{(*E*)-[2-(4-nitrophenyl)hydrazinylidene]methyl}-1*H*-pyrazole-1,3-diyl)dibenzoic acid (29).** Reddish solid (372 mg, 79%). ^1^H NMR, 300 MHz (DMSO-*d*_6_): δ 11.31 (s, 1H), 9.17 (s, 1H), 8.15–8.08 (m, 9H), 7.89 (d, *J* = 8.0 Hz, 2H), 7.10 (d, *J* = 8.4 Hz, 2H); ^13^C NMR (75 MHz, DMSO-*d*_6_) δ = 167.7, 167.2, 151.0, 150.9, 142.2, 138.5, 136.5, 134.6, 131.6, 131.3, 130.0, 129.8, 128.8, 128.3, 126.5, 118.9, 118.7, 111.5. HRMS (ESI-FTMS Mass (m/z): calcd for C_24_H_17_N_5_O_6_ [M + H]^+^ = 472.1252, found 472.1250.

All the Spectra can be seen in the Appendix A.

### 2.2. In Vitro and In Vivo Toxicity

Synthesized molecules were tested for their possible toxicity to human cell lines. These compounds did not show any noticeable growth inhibition for the NCI-60 cancer cell lines at 10 μM concentration. Furthermore, compounds showing activity against *A. baumannii* were tested for their possible toxicity against human embryonic kidney (HEK293) cell line and they did not show any significant toxicity up to 50 µg/mL concentration. 

After finding the benign nature of potent compounds against human cell lines, we tested the most potent compound **18** for any toxic in vivo effect on mice (Figure 2). We chose the doses of 20 and 50 mg/kg, which were used in our previous studies [20]. In vivo effects of a single IP injection of the compound were assessed by 14 different parameters for organs’ functions as described in Methods section and shown in Figure 2. This measurement clearly showed that none of the organ function markers indicated a toxicity by the used criteria. The majority of blood tests after the compound **18** administration showed no significant difference from control samples, and all except one of them were within the normal ranges. In particular, a normal concentration of albumin indicated no harm to the liver and kidneys. Unaffected blood urea nitrogen, creatinine, sodium, potassium levels indicated no harm to the kidneys of the treated animals. A normal concentration of amylase indicated that this potent anti-*A baumannii* agent did not adversely affect the pancreas. A normal alkaline phosphatase (ALP) level in blood is a key indicator of a healthy liver and bones. This compound treatment showed a slightly lower level of ALP but within the normal range. Normal alanine transaminase (ALT), and glucose levels at 20 mg/kg treatment further confirm the tolerance of this compound by the liver. Although a slight elevation of glucose was observed at 50 mg/kg dose, liver injury was not confirmed by other liver injury markers, such as total bilirubin, ALP, and ALT. Glucose level is very labile and it changes constantly during the day depending, for example, on food consumption. Calcium and phosphorus levels indicated the normal function of several organs such as liver, kidney, and bones. Unaffected total protein and globulin levels indicate healthy liver and kidney, and immune system respectively. Thus, this narrow-spectrum antibiotic is non-toxic in therapeutic doses, and seems to be very safe for further drug development. Future studies may need to focus on the development of a final pharmaceutical product. For this, additional tests would be needed to detect pharmacological responses through absorption, distribution, metabolism and excretion studies. All those will need to be done prior to Phase I testing in humans.

### 2.3. Calculated Physicochemical Properties 

We calculated the physicochemical properties of the most potent compound **18** by using a free online software, SwissADME (http://www.swissadme.ch/index.php). The n-octanol/water partition coefficient (ilog*P*) value for this potent molecule is 2.67, which is within the optimum range [21]. The Topological Total Surface Area (TPSA) is 116.81 A°, which should allow good passive transport across the cell membrane. There is no violation of Lipinski’s rule of five and zero alert for PAINS. All these favorable parameters and very good biological activities indicate the suitability of the molecule **18** for further antibiotic development.

## 3. Materials and Methods 

### General Consideration

All the products were obtained by reactions carried out in round-bottom flasks under an air atmosphere. Reagents, solvents, and substrates were purchased from Oakwood Chemical (Estill, SC, USA) and Fisher Scientific (Hanover Park, IL, USA.). ^1^H and ^13^C spectra were recorded with a Varian Mercury −300 MHz and 75 MHz respectively in DMSO-*d*_6_ solvent with TMS as internal standard. ESI-FTMS mass spectra were recorded in Brucker Apex II-FTMS system. Growth media and bacterial broth were purchased from Fisher Scientific, ATCC, and ATCCHardy Diagnostics (Santa Maria, CA, USA). 

Synthesis of pyrazole aldehyde (**4**): A mixture of 4-actylbenzoic acid (**2**, 1.64 g, 10 mmol) and 4-hydrazinobenzoic acid (**1**, 1.980 g, 10.5 mmol) in ethanol was refluxed for 8 h and ethanol was removed by evaporation following drying in vacuo to get the hydrazone intermediate (**3**), which was subjected to further reaction without purification. The hydrazone derivative (**3**) was dissolved in *N*,*N*-dimethylformamide (30 mL), and cooled under ice for ~15 min followed by the dropwise addition of phosphorous oxychloride (POCl_3_, 4.67 mL, 50 mmol). The reaction mixture was stirred under ice for 30 min, and heated to 80 °C for 12 h. The reaction mixture was poured onto ice and the aqueous mixture was stirred for 12 h. The solid product was filtered by gravity filtration and washed repeatedly with water followed by drying under vacuum to get the pure product (**4**). 

Synthesis of hydrazones (**4**–**29**): A mixture of aldehyde (**4**, 336 mg, 1 mmol) and the hydrazine derivative (1.05 mmol) and sodium acetate (86 mg, 1.05 mmol) in case of the hydrochloride salt of the hydrazine derivatives in anhydrous ethanol was refluxed for 8 h. Water was added in the mixture and the solid product was filtered followed by washing with water and ethanol to get the pure products for biological studies.

Culturing of Bacteria: Tryptic soy agar (TSA) slants were used to maintain bacterial cultures. Cation-Adjusted Mueller Hinton Broth (CAMHB) was used to grow bacteria in liquid culture including 96-well plates for minimum inhibition concentration (MIC) testing and phosphate-buffered saline (PBS) was used to make bacterial dilutions.

MIC studies: The MICs of potent compounds were determined by using the broth microdilution method following the Clinical and Laboratory Standards Institute (CLSI) guidelines. The starting concentration of compounds was 50 μg/mL with two-fold serial dilutions below for MIC determination. The MIC values were measured in three independent experiments on different days with fresh bacterial culture and confirmed with at least duplicate occurrence. The concentration of compound in diluent dimethyl sulfoxide (DMSO) was maintained at 2.5%, which is below the cytotoxicity level for bacteria.

In vitro Cytotoxicity Studies: Cytotoxicity of compounds against human embryonic (HEK293) kidney cell line and NCI-60 cancer cell lines were performed in 96-well black plate using resazurin cell viability assay. Cells (4000 per well) were plated in Eagle’s Minimum Essential Medium (EMEM) with 10% Fetal Bovine Serum (FBS) and incubated at 37 °C in the presence of 5% carbon dioxide for 24 h. After incubation, test compounds at different concentrations were added to each well and placed in 96-well plate shaker for 5 min for proper mixing of compounds. The plate was re-incubated for 24 h. After the second incubation, 40 µL resazurin (0.15 mg/mL, w/v) was added in each well and mixed gently by pipetting. The plate was incubated for 4 additional hours and plate reading was conducted using Bio Tek^TM^ Cytation^TM^5 plate reader with excitation at 560 nm and emission at 590 nm. Data were processed in Microsoft Excel^®^ for Office 365 MSO.

In Vivo Toxicity Studies: All animal experiments were performed at the Central Arkansas Veterans Healthcare System (John L. McClellan Memorial Veterans Hospital in Little Rock, AR) and have been approved by the Institutional Animal Care and Use Committee. CD-1 male mice (8 weeks old, 33–37 g) were purchased from Charles River Laboratories (Wilmington, MA). The test compound **18** was freshly dissolved in 0.9% saline, sterilized by ultrafiltration, and injected intraperitoneally (IP) in mice at two doses of 20 or 50 mg/kg (*n* = 5 per dose). The two additional control groups (n = 5/group) were untreated or administered with the vehicle (saline). The mice were euthanized 24 h after the injection, and blood was collected by cardiac puncture. Toxicity was assessed by measuring 14 blood markers of various organ function available in the Comprehensive Diagnosis Kit (Abaxis, Union City, CA, USA) and using VetScan VS2 instrument (Abaxis). The markers included: alanine aminotransferase (ALT), albumin (ALB), alkaline phosphatase (ALP), amylase (AMY), calcium (CA), creatinine (CRE), globulin (GLOB), glucose (GLU), phosphorus (PHOS), potassium (K^+^), sodium (NA^+^), total bilirubin (TBIL), total protein (TP), and blood urea nitrogen (BUN). Our criterium for toxicity was the combination of: (a) measurements being beyond the normal ranges, (b) statistically significantly difference from the untreated and vehicle (saline) controls, and (c) consistency between several functional markers of the same organ.

## 4. Conclusions

We reported the synthesis of 25 novel pyrazole compounds. They were synthesized efficiently by using readily available starting materials and benign conditions. Column purification was not required to get pure products. The molecules were tested against both Gram-positive and Gram-negative bacteria and found to be potent and specific inhibitors of *A. baumannii* at low concentrations. These molecules are non-toxic in in vitro and in vivo studies. Thus, these potent compounds with very good selectivity factor could be pursued for further drug development as narrow-spectrum antibiotics. Mode of action and further antimicrobial studies are going on and will be reported elsewhere. 

## 5. Patents

Alam, M. A. Antimicrobial agents and the method of synthesizing the antimicrobial agents.

US Patent. 10,596,153, 2020.

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
