# Peer review of "Synthesis of 4,4′-(4-Formyl-1H-pyrazole-1,3-diyl)dibenzoic Acid Derivatives as Narrow Spectrum Antibiotics for the Potential Treatment of Acinetobacter Baumannii Infections"

_antibiotics, 2020, doi:10.3390/antibiotics9100650_

Round 1

Reviewer 1 Report

This manuscript describes the synthesis of 25 pyrazole-derived hydrazine congeners and their antimicrobial activity against Gram-positive bacteria.  In addition, toxicities of the synthesized compounds were evaluated in vitro and vivo.  I think there are several interesting results in this manuscript.  However, antimicrobial activity of the synthesized compounds was not so potent, and structure activity relationship studies are poor.  Therefore, I recommend the authors to submit to another suitable journal.  If this manuscript would be revised to publish in antibiotics, I would like to say some comments as below.

The authors should explain the reason why they designed these 25 pyrazole-derived hydrazine compounds.  It is described that halogen-substituted N-phenyl hydrazone showed potent activity.  However, the synthesized halogen-substituted N-phenyl hydrazones were limited.  Why did not the author synthesize 4-fluorophenyl, 2-chlorophenyl, 3-chlorophenyl, and so on substituted compounds?

Why did the author use colistin as a positive control?  Is this the best choice?

There are no MS data in this manuscript.  Don’t the authors impose to measure MS data in this journal?

I suggest that the authors would add NMR data as supplement data to prove the purity of the synthesized compounds.

Author Response

Reviewer 1

This manuscript describes the synthesis of 25 pyrazole-derived hydrazine congeners and their antimicrobial activity against Gram-positive bacteria.  In addition, toxicities of the synthesized compounds were evaluated in vitro and vivo.  I think there are several interesting results in this manuscript.  However, antimicrobial activity of the synthesized compounds was not so potent, and structure activity relationship studies are poor.  Therefore, I recommend the authors to submit to another suitable journal.  If this manuscript would be revised to publish in antibiotics, I would like to say some comments as below.

The authors should explain the reason why they designed these 25 pyrazole-derived hydrazine compounds.  It is described that halogen-substituted N-phenyl hydrazone showed potent activity.  However, the synthesized halogen-substituted N-phenyl hydrazones were limited.  Why did not the author synthesize 4-fluorophenyl, 2-chlorophenyl, 3-chlorophenyl, and so on substituted compounds?

We appreciate this important comment of the reviewer. We synthesized several other halogen substituted phenyl derivatives. However, we could not obtain >95% purity for many of them. Therefore, we did not include those compounds.

Why did the author use colistin as a positive control?  Is this the best choice?

As these molecules are potent against A. baumannii, so we used colistin as a positive control. We have also used vancomycin against Gram-positive strains, but we did not include values in the manuscript, as our compounds did not show any reasonable activity against Gram-positive strains.

There are no MS data in this manuscript.  Don’t the authors impose to measure MS data in this journal?

We included the MS data for the compounds in the manuscript.

I suggest that the authors would add NMR data as supplement data to prove the purity of the synthesized compounds.

Supplementary Information has been added.

Reviewer 2 Report

The manuscript entitled” Synthesis of 4,4'-(4-Formyl-1H-pyrazole-1,3-diyl)dibenzoic Acid 
Derivatives as Narrow Spectrum Antibiotics for Potential Treatment of Acinetobacter baumannii Infections” deals with the synthesis and evaluation of antibacterial activity of synthesized compounds against Acinetobacter baumannii. I have an impression that manuscript is written casually. Besides, significant plagiarism was detected. Actually all Introduction have significant plagiarism.

Major:

In Introduction authors gave figure 1 which does not mention in the text and it is not clear what for it is.

Page 2. In the process of synthesis of compound 4 in Scheme 1 , what is written on arrows corresponds   is written in page 7 line 160, whwre authors describe the synthesis of compound 4.

Page 3.Line 107. The R for compound 5 is missing in the table.

Since all compounds are crystals it is necessary to give their melting point which is very important physico-chemical characteristic of compounds.

Page 7. Line 157.It is written” ESI-FTMS mass spectra were recorded in Brucker Apex II-FTMS system”,  but in experimental part there are no data. Furthermore, authors did not give data on elemental analysis. So, there are no brutto formulas of compounds and this make difficult the check of NMR data.

There is no description of NMR spectra. Authors wrote only the number of protons, without giving the description as usual required. For this reason, probably in all NMR spectra some protons are missing. For example.

Page 8. Line 21 . Compound 4. C18H12N2O5 according to the formula has 12 protons but authors mentioned 10.

Page 11.Line 338. Compound 29 according to the formula C24H17N5O6 has 17 protons , authors mentioned 14.

IR spectra are missing.

Page 11. Conclusion is not complete.

Furthermore, there are many mistakes

Page 2. Line 35. Should be: …in communities [2-5]

Page 2. Line 36. Should be:    War began in 2003 [6]

Page 2. Line 38. Should be:…community [7-9].

Page 2. Line 40. Should be:… the top of the list [10-11]

Page 2. Line 48. Should be:… multidrug-resistance infections [12].I

Page 2. Line 55. Should be:  A. baumannii agents [14-15]

Page 2 Line 56. Should be:… significantly [16-17]

Page 2. Line 58. Should be:.. particularly S. aureus [18-20]

Page 2. Line 63. Should be: hydrazinobenzoic acid (2) with 4-acetylbenzoic acid (1) …

Page 3. Line 103. The abbreviation of bacteria strains should be below the table and not in the title.

Page 5. Line 117. Should be: .. in previous studies [20]

Page 5.Line 120: Compound 18 should be in bold.

Page 6. Line 139. The title of the figure usually paced below figure and not before.

Page 6. Line 139. Compounds 18 should be in bold. In figure 2 there is no abbreviation for U and S.

Page 7. Lines 145 and 151: Compound 18 should be in bold.

Page 7. Line 148.Should be:  range [21].

Page 7. Line 160. Should be: … 4-acetylbenzoic acid ..

Page 7. Line 164.Better to say:   in N,N-dimethylformamide (30 mL)

Page 11. Line 363 Authors in acknowledgement mention Mass spectrometry data which do not exist in experimental part.

Page 11. Line 374. Should be: Acinetobacter baumannii:

Page 11. Line 377 : Should be: Acinetobacter baumannii

Author Response

Reviewer 2

The manuscript entitled” Synthesis of 4,4'-(4-Formyl-1H-pyrazole-1,3-diyl)dibenzoic Acid 

Derivatives as Narrow Spectrum Antibiotics for Potential Treatment of Acinetobacter baumannii Infections” deals with the synthesis and evaluation of antibacterial activity of synthesized compounds against Acinetobacter baumannii. I have an impression that manuscript is written casually. Besides, significant plagiarism was detected. Actually all Introduction have significant plagiarism.

The introduction part has been paraphrased to remove the plagiarism.

Major:

In Introduction authors gave figure 1 which does not mention in the text and it is not clear what for it is.

We mentioned Figure 1 in the test.

Page 2. In the process of synthesis of compound 4 in Scheme 1 , what is written on arrows corresponds   is written in page 7 line 160, whwre authors describe the synthesis of compound 4.

Sorry, we could not understand this comment. We double-checked the section and we found that they are correct.

Page 3.Line 107. The R for compound 5 is missing in the table.

R is N,N-dimethyl in the table.

Since all compounds are crystals it is necessary to give their melting point which is very important physico-chemical characteristic of compounds.

Compounds are amorphous solid. Therefore, we are not reporting their melting point.

Page 7. Line 157.It is written” ESI-FTMS mass spectra were recorded in Brucker Apex II-FTMS system”,  but in experimental part there are no data. Furthermore, authors did not give data on elemental analysis. So, there are no brutto formulas of compounds and this make difficult the check of NMR data.

We could not attach the supporting information file during the submission. Now the SI is attached and we have added HRMS data in the manuscript.

There is no description of NMR spectra. Authors wrote only the number of protons, without giving the description as usual required. For this reason, probably in all NMR spectra some protons are missing. For example.

We appreciate this comment. We have found that NH and CO2H protons are missing in most of our 1H NMR spectra. These protons may merge with the H2O due to rapid exchange.

Page 8. Line 21 . Compound 4. C18H12N2O5 according to the formula has 12 protons but authors mentioned 10.

We did not observe the CO2H protons ~12 ppm. These protons may be merging with the water peak.

Page 11.Line 338. Compound 29 according to the formula C24H17N5O6 has 17 protons , authors mentioned 14.

Both the spectra, 1H and 13C, for this compound is very clean and three labile hydrogens are missing, may be merging with the solvent at ~3.5 ppm.

IR spectra are missing.

We believe NMR and HRMS data are enough to characterize these simple molecules.

Page 11. Conclusion is not complete.

We expanded the conclusion section.

Furthermore, there are many mistakes

We appreciate the reviewer for editing the manuscript so minutely. We made the following and similar changes in the manuscript.

Page 2. Line 35. Should be: …in communities [2-5]

Page 2. Line 36. Should be:    War began in 2003 [6]

Page 2. Line 38. Should be:…community [7-9].

Page 2. Line 40. Should be:… the top of the list [10-11]

Page 2. Line 48. Should be:… multidrug-resistance infections [12].I

Page 2. Line 55. Should be:  A. baumannii agents [14-15]

Page 2 Line 56. Should be:… significantly [16-17]

Page 2. Line 58. Should be:.. particularly S. aureus [18-20]

Page 2. Line 63. Should be: hydrazinobenzoic acid (2) with 4-acetylbenzoic acid (1) …

Page 3. Line 103. The abbreviation of bacteria strains should be below the table and not in the title.

Page 5. Line 117. Should be: .. in previous studies [20]

Page 5.Line 120: Compound 18 should be in bold.

Page 6. Line 139. The title of the figure usually paced below figure and not before.

Page 6. Line 139. Compounds 18 should be in bold. In figure 2 there is no abbreviation for U and S.

Page 7. Lines 145 and 151: Compound 18 should be in bold.

Page 7. Line 148.Should be:  range [21].

Page 7. Line 160. Should be: … 4-acetylbenzoic acid ..

Page 7. Line 164.Better to say:   in N,N-dimethylformamide (30 mL)

Page 11. Line 363 Authors in acknowledgement mention Mass spectrometry data which do not exist in experimental part.

Page 11. Line 374. Should be: Acinetobacter baumannii:

Page 11. Line 377 : Should be: Acinetobacter baumannii

Reviewer 3 Report

Dear Authors,

Line 346-327: Please correct to “Gram-positive and Gram-negative”

Author Response

Line 346-327: Please correct to “Gram-positive and Gram-negative”

We made the change in the manuscript.

Reviewer 4 Report

Dear authors, please consider addressing the following issues:

  1. Line 102 “coumarin” ?
  2. Please provide Mass spectra fir the new molecules, and upload ALL (including CNMR, HNMR) spectra as supplementary materials.
  3. Some paragraphs are identical with previous works published by this research group. Try to rephrase or just refence your work without repeating all the information. Line 196-208

Author Response

Line 102 “coumarin” ?

We removed ‘’coumarin.’’

Please provide Mass spectra fir the new molecules, and upload ALL (including CNMR, HNMR) spectra as supplementary materials.

We added HRMS data and uploaded the supporting material file containing HNMR and CNMR spectra.

Some paragraphs are identical with previous works published by this research group. Try to rephrase or just refence your work without repeating all the information. Line 196-208.

We made the changes and highlighted the information in the text.

Round 2

Reviewer 1 Report

I accept your effort.

Author Response

Page 8. Compound 4. Compound 4. C18H12N2O5 according to the formula has 12 protons but authors mentioned 10.

We appreciate the reviewer for this comment. Two missing protons are from two carboxylic acid functional group, which may be in the offset region or merging with H2O (DMSO-d6) protons.

Page 11.Line 338. Compound 29 according to the formula C24H17N5O6 has 17 protons , authors mentioned 14.

We thank the reviewer for finding out this error. We added one more proton at d 8.15-8.08 (m, 9H). Now two less proton are from the two carboxylic acid functional group, which may merge with the H2O protons.

IR spectra are missing.

We have thoroughly characterized these new molecules by 1H and 13C NMR spectroscopy and confirmed their structures by HRMS. IR spectra would have been better but we believe that we are not missing anything related to the structure of molecules without IR data.

Page 6. Line 139. The title of the figure usually paced below figure and not before. On page 7 authors mentioned : 4.Material and methods. On line 162. Experimental procedure

We move the title of the figure below figure. We removed the Experimental procedure from line 162.

In page 11. Again is mentioned :Experimental data. Actually the description of compounds should be in page 7, starting from line 177. And after finishing with description of compounds should follow biological experiment.

We move the experimental data after line 177.

Reviewer 2 Report

Page 8. Compound 4. Compound 4. C18H12N2O5 according to the formula has 12 protons but authors mentioned 10.

Page 11.Line 338. Compound 29 according to the formula C24H17N5O6 has 17 protons , authors mentioned 14.

IR spectra are missing.

Page 6. Line 139. The title of the figure usually paced below figure and not before.On page 7 authors mentioned :4.Material and methods. On line 162. Experimental procedure

In page 11. Again is mentioned :Experimental data. Actually the description of compounds should be in page 7, starting from line 177. And after finishing with description of compounds should follow biological experiment.

Author Response

(The authors gave the same response as above.)

Reviewer 4 Report

Thank you for taking into consideration my suggestions.  

Author Response

We thank you for your insightful comments and time.